# A Heuristic Approach to Analysis of the Genetic Susceptibility Profile in Patients Affected by Airway Allergies

**DOI:** 10.3390/genes15081105

**Published:** 2024-08-22

**Authors:** Domenico Lio, Gabriele Di Lorenzo, Ignazio Brusca, Letizia Scola, Chiara Bellia, Simona La Piana, Maria Barrale, Manuela Bova, Loredana Vaccarino, Giusi Irma Forte, Giovanni Pilato

**Affiliations:** 1University Research Center “Migrate”, University of Palermo, 90100 Palermo, Italy; 2Department of Health Promotion Sciences, Maternal and Infant Care, Internal Medicine and Medical Specialties (PROMISE), University of Palermo, 90100 Palermo, Italy; gabriele.dilorenzo@unipa.it (G.D.L.); simonalapiana@gmail.com (S.L.P.); 3Clinical Pathology Unit, Buccheri La Ferla Hospital of Palermo, 90100 Palermo, Italy; brusca.ignazio@fbfpa.it (I.B.); barrale.maria@fbfpa.it (M.B.); 4Clinical Pathology, Department of Bio-Medicine, Neuroscience and Advanced Diagnostics, University of Palermo, 90100 Palermo, Italy; letizia.scola@unipa.it (L.S.); manuelabova@hotmail.it (M.B.); loredanavaccarino@libero.it (L.V.); 5Transfusion Medicine Unit, University Hospital “Paolo Giaccone”, 90100 Palermo, Italy; chiara.bellia@unipa.it; 6Department of Biomedicine, Neurosciences and Advanced Diagnostics, University of Palermo, 90100 Palermo, Italy; 7Institute of Molecular Bioimaging and Physiology (IBFM), National Research Council (CNR), Cefalù Secondary Site, C/da Pietrapollastra-Pisciotto, 90015 Cefalù, Italy; giusi.forte@ibfm.cnr.it; 8Institute for High Performance Computing and Networking (ICAR), National Research Council (CNR), 90100 Palermo, Italy; giovanni.pilato@icar.cnr.it

**Keywords:** allergy, airways, cytokine gene SNPs, Klotho, heuristic approach, market basket analysis

## Abstract

Allergic respiratory diseases such as asthma might be considered multifactorial diseases, having a complex pathogenesis that involves environmental factors and the activation of a large set of immune response pathways and mechanisms. In addition, variations in genetic background seem to play a central role. The method developed for the analysis of the complexities, as association rule mining, nowadays may be applied to different research areas including genetic and biological complexities such as atopic airway diseases to identify complex genetic or biological markers and enlighten new diagnostic and therapeutic targets. A total of 308 allergic patients and 205 controls were typed for 13 single nucleotide polymorphisms (SNPs) of cytokine and receptors genes involved in type 1 and type 2 inflammatory response (IL-4 rs2243250 C/T, IL-4R rs1801275A/G, IL-6 rs1800795 G/C, IL-10 rs1800872 A/C and rs1800896 A/G, IL-10RB rs2834167A/G, IL-13 rs1800925 C/T, IL-18 rs187238G/C, IFNγ rs 24030561A/T and IFNγR2 rs2834213G/A), the rs2228137C/T of CD23 receptor gene and rs577912C/T and rs564481C/T of Klotho genes, using KASPar SNP genotyping method. Clinical and laboratory data of patients were analyzed by formal statistic tools and by a data-mining technique—market basket analysis—selecting a minimum threshold of 90% of rule confidence. Formal statistical analyses show that IL-6 rs1800795GG, IL-10RB rs2834167G positive genotypes, IL-13 rs1800925CC, CD23 rs2228137TT Klotho rs564481TT, might be risk factors for allergy. Applying the association rule methodology, we identify 10 genotype combination patterns associated with susceptibility to allergies. Together these data necessitate being confirmed in further studies, indicating that the heuristic approach might be a straightforward and useful tool to find predictive and diagnostic molecular patterns that might be also considered potential therapeutic targets in allergy.

## 1. Introduction

Allergic diseases have an increased diffusion, with a major role for airways allergic pathologies, such as rhinitis or asthma, which currently affect up to 400 million people worldwide [1]. As is well known, the onset of allergic airway diseases is influenced by the modification of the immune system balance. In particular, predisposition to the production of Th2 cytokines (i.e., IL-4 and IL-13, essential for IgE production [2] plays a central role. In addition, reduced stimulation of immune response (Th1 and TH17) towards microbes colonizing the mucosal environment, as well as a reduced T-reg cell-mediated suppression, contribute to the hyper-responsiveness against allergens [3].

On the other hand, the imbalance between Th1 and Th2 cells does not explain all of this complex phenomenon. Evidence is cumulating on the role of Th1-mediated inflammation (type 1 inflammation) [4,5]. Type 1 inflammation may play a role in maintaining respiratory mucosal hypersensitivity due to chronic reactivation of microbial infections [4,5].

It has long been known that genetic predisposition plays a fundamental role in the appearance of these pathologies. Actually, variations of genes encoding for molecules involved in the pathogenesis of allergic diseases as cytokine networks regulating IgE switching production and mechanisms of allergic reaction triggering and maintaining, appear the most likely candidates for genetic predisposition to allergic disease and Atopy [6]. In particular, genetic polymorphisms of the CD23 gene (FcεR2, encoding the low-affinity IgE receptor (FcεRII), interleukin(IL-)4, IL-13, and related receptors, regulating the Type 2 immune response, have been frequently found associated to atopy and allergic diseases [7]. However, other gene polymorphisms of Type 1 and inflammation network cytokines have been studied [7]. In particular, IFN-γ gene variants have been analyzed as possible markers of predisposition to allergy [8].

Moreover, data are emerging on the role that the klotho gene can play in the mechanisms involved in the inhibition of the allergic response activation. α-Klotho (KL) is an anti-aging protein and has been shown to exert anti-inflammatory and anti-oxidative effects in lung and pulmonary diseases [9]. More recently, Brandao-Rangel et al. reported that stimulation of α-Klotho production may have positive effects on the prevention of asthmatic symptoms [10].

In spite of the extended studies devoted to the identification of single variants of candidate genes, this approach appears to be inadequate as allergic diseases are multifactorial diseases in which the single gene effect might be only partial or marginal and might be different in different moments of a diseases’ clinical development [11]. In addition, genome-wide studies do not produce clear results useful in the prediction of allergic risk and prevention [12]. Non-Mendelian forms of inheritance, gene–gene interactions, or analyses of functional pathways might be more informative [13]. Several tools are being used to detect significant associations. One of the most promising approaches is the use of data mining algorithms that can highlight gene interactions and allele clusters that simultaneously influence multiple metabolic pathways directly and indirectly linked to the predisposition to allergies [14]. This kind of analysis, defined also as heuristic analysis, uses a set of mathematical techniques, and procedures to reach a strategy appropriate to solving a given problem, as the relevance of the simultaneous presence of polymorphisms of genes associated with predisposition to a disease [15]. In this view, we apply an a priori algorithm, named market basket analysis [16] to a set of data, obtained by typing more than 500 subjects for a set of 13 SNPs (selected on the basis of their functional effects on the gene products, Table 1), in an attempt to identify different multi-locus associations that have an high probability to predict the genetical susceptibility to the respiratory allergy.

## 2. Materials and Methods

### 2.1. Patients and Controls

Allergic patients and controls were recruited for the study from March 2015 to December 2018. Patients group, 308 subjects aged 18 years and over, born in West Sicily was recruited among subjects presenting with a history of respiratory allergy to the Outpatient Allergy Clinic of the Medical Department of the University of Palermo, Italy, and U.O.C of Clinical Pathology Buccheri La Ferla Hospital of Palermo, Italy. Clinical examination was performed as previously described [28,29]. Briefly, all patients were diagnosed with allergic rhinitis and/or asthma based on patient-reported symptoms, physical examination (rhinoscopy and spirometry), and showed a normal lung function test (baseline FEV1 ≥ 80% of predictive value). All patients have positive skin test responses for 1 or more of the following common airborne allergens (positive skin prick test): Phleum pratense, Lolium perenne, Poa pratensis, Artemisia vulgaris, Parietaria judaica, Plantago lanceolata, olive, birch, hazel, oak, cypress, plane trees, Dermatophagoides pteronyssinus, and farinae, dog, cat, and horse dander, Alternaria alternata, Cladosporium herbarum, Aspergillus mixture, latex, and cockroach. Patients affected by autoimmune diseases, neoplastic kidney, cardiovascular, or BPC pathologies were excluded from the study. At the time of diagnosis, blood was collected for analysis of serum t-IgE levels and genetic analyses. Family history of allergy, as well as history of smoking, were recorded. The control group comprised 205 adult apparently healthy volunteers who self-report to be negative for respiratory allergies as well as chronic urticarial diseases, atopic dermatitis, or vasomotor rhinitis. A general clinical examination and a thorough anamnesis regarding family history of allergies and evaluation of recreational habits (smoking and/or alcohol) were performed. Family history of allergy and presence of chronic diseases were exclusion criteria. Demographic and clinical data are detailed in Table 2.

Our study was performed in accordance with ethical standards of the Helsinki Declaration of the World Medical Association and Italian legislation and was approved by the local institutional review board (Comitato Etico Palermo 1, protocol code CET21, date of approval 23 February 2015). All participants gave their informed consent, and the data were encoded to ensure the protection of patients and control privacy.

### 2.2. SNP Genotyping SNP Molecular Typing

The EDTA peripheral blood samples used for molecular typing, respectively, were collected and stored at 70 °C until DNA extraction. We selected thirteen functional and common SNPs on the dbSNP NCBI database (http://www.ensembl.org/index.html, last access: 4 April 2024). DNA samples were typed for 13 single nucleotide polymorphisms (SNPs) of cytokine and receptors genes involved in type 1 and type 2 inflammatory response (Table 1), using a competitive allele-specific PCR (polymerase chain reaction) assays (KASPar), developed by K-Bioscience (K-Bioscience Ltd., Hoddesdon, UK), as previously described [23] with an accuracy of 99.8% (https://www.biosearchtech.com/products/oligos-probes-and-primers/kasp-genotyping-assays, last access: 7 August 2024). Genotypes were revealed using the 7300 system SDS software, version 1.3 (Applera Italia, MONZA (MI), Italy), able to as previously described [23] with an accuracy of >99.8 [30].

### 2.3. Statistical Analysis

SNP allele and genotype frequency evaluation by gene count using an online statistical analysis tool (https://www.snpstats.net/start.htm, last access: 2 April 2024) This online tool allows to easily perform descriptive analysis, test for Hardy–Weinberg equilibrium, analysis of single SNPs or multiple SNPs inheritance models, and analysis of interactions (gene–gene or gene–environment) [31]. Pearson’s test was applied to test Hardy–Weinberg equilibrium to check the absence of non-random mating or genetic drift in the population samples recruited. Power of calculation has been evaluated for all comparisons made using an online tool (https://sample-size.net/, last access: 1 December 2023) to establish the sample size necessary to identify a real difference among the groups from that obtained by chance. This study had 80.0% power to detect a P1 = 0.066 with a risk ratio = 0.220 and an odds ratio = 0.165 with a minimum sample size of 113 allergic subjects and 75 control subjects for allergy. For the less frequent SNP genotype, the minimum sample size was 320 allergic subjects and 190 control subjects.

Significant differences in allele, homozygous, and heterozygous genotype distributions were analyzed using Fisher’s exact test (adjusted for age and gender). Multiple logistic regression models were applied (snpstats tool), using dominant (wild allele homozygotes versus heterozygotes plus minor allele homozygotes) and recessive (minor allele homozygotes plus heterozygotes versus wild allele homozygotes) models for each SNP typed. Age, gender, and the other SNPs analyzed were included as covariate factors, and odds ratios (OR), 95% confidence intervals (95% C.I.), and *p*-values (*p*-value cutoff < 0.05) were adjusted accordingly.

### 2.4. Heuristic Analysis

A heuristic approach has been applied to analysis of the dataset containing both patient and control subjects to identify patterns that may indicate the presence of allergy in a subject. 

The items in the dataset were analyzed considering the allergy target variable (values: Yes/No). The features considered for each row (referred to as *record*) are gender, age, and the thirteen SNPs typed. The dataset was analyzed using a market basket analysis approach through an efficient and well-tested implementation of the a priori algorithm proposed by Agrawal et al. in 1994 [16,32]. Algorithm procedure is described in Figure 1.

As described in Figure 1, the dataset records are first transformed into *itemsets* (i.e., sets of items). This step is necessary to apply the a priori algorithm. In particular, each value in the table of the dataset is encoded as a pair (“*Feature Name*”, “*Feature Value*”), which constitutes an *item* of the itemset. The a priori algorithm is then applied to the list of itemsets to extract a set of rules [33].

More formally, let *I = {i*_1_, *i*_2_, …, *i_n_}* be the set of all the *n* pairs in the form (“*FeatureName*”, “*FeatureValue*”) encoded from the original dataset. We refer to each pair as an *item*. Let *T = {t*_1_, *t*_2_, …, *t_m_}* be a set of transactions that characterize each record of the original dataset; *m* is the number of rows in the original dataset. Each entry in *T* has a unique identifier and contains a subset of the items in *I*.

A rule is defined as an implication of the form X ⇒ Y where X, Y ⊆ *I* and X ∩ Y = ∅. The sets X and Y are named *antecedent* (or left-hand-side, LHS) and *consequent* (or right-hand-side, RHS) of the rule, respectively.

In our case, the items of *I* are the pairs (“*Feature Name*”, “*Feature Value*”) and the consequent that we focus on is (“*Allergic*”, “*Yes*”).

The efficient a priori algorithm is then run on *T*, producing a set of rules. For each rule found, the procedure computes a set of four well-known measures of significance and interest: *confidence*, *support*, *lift*, and *conviction*. Their meaning is briefly recalled in the following.

Given a rule of the form X → Y, the *confidence* is the probability that the right part of the rule is given by the left part of the rule. If X → Y, the confidence is P(Y|X). From a computational point of view, it is the ratio between the number of data strings containing both X and Y and the number of data strings containing X.

The *support* is the frequency with which a rule’s left and right parts appear together in the dataset. If X → Y, the support is the joint probability P(Y, X); from a computational point of view, it is the ratio between the number of data strings containing both X and Y and the total number of data strings.

The *lift* is the ratio of observed support to expected support. If X → Y, then the lift is given by the fraction P(X, Y)/(P(X) * P(Y)); from a computational point of view, it is the ratio between the number of transactions containing both X and Y and the product of the transactions containing X and the fraction of transactions containing Y.

The *conviction* is the ratio P(-Y)/P(-Y|X), i.e., the ratio of the frequency with which Y does not appear in the data to the frequency with which Y does not appear in the data given X. A high value of conviction denotes strong dependency between X and Y, suggesting that Y is unlikely to happen without X being present.

Among the rules generated by the efficient a priori algorithm, only those rules having (“*Allergic*”, ”*Yes*”) as consequent and that is associated with a support value larger than 0.1 (i.e., 10%) and a confidence value larger than 0.9 (i.e., 90%), were selected.

GraphPad InStat software version 3.06 (GraphPad, San Diego, CA, USA) was used to determine performances of the rules identified by heuristic analysis, in terms of sensitivity (i.e., probability that an affected patient might be identified the test), specificity (probability that an unaffected subject might be identified the test), positive predictive value (PPV, probability that a subject positive for the test might result really affected), negative predictive value (NPV, probability that a subject negative for the test might be unaffected).

## 3. Results

Table 3 shows the allele and genotype frequencies of all SNP types. Comparison of allelic and genotypic frequencies between the two groups highlighted an increase in the frequency of positive genotypes for the T allele of the SNP rs2228137 of the gene encoding the type 2 receptor for IgE (CD23) and an increase in the frequency of the rs1800925CC genotype of the IL13 gene. IL-6 rs1800795G allele (in particular GG genotype) was increased in the allergy patient group as well as IL-10 rs1800896AA genotype and IL-10RB rs2834167G/* positive genotypes. Finally, the Klotho rs564481 TT genotype was increased in allergy.

Using the market basket analysis approach (see Section 2), ten rules characterized by (“*Allergic*”, “*Yes*”) as “consequent” associated with a “support” value ≥ 0.1 (i.e., 10%) and a “confidence” value ≥ 0.9 (i.e., 90%) (Table 4). These rules highlight clusters of genotypes that have a 90% probability of identifying a subject predisposed to allergy. Nine of these were characterized by the presence of the IL-10RB rs2834167G/G genotype, in five the IL-4 rs2243250C/C genotype was present in two, IL-4R rs1801275A/A in two, IL-13 rs1800925C/C in three, IL-6 rs1800795G/G in two, Klotho rs577912C/C and another Klotho rs564481T/T in two, respectively; finally, one of the clusters contains the IFNγR2 rs2834213G/G genotype.

The differences in the distribution of the clusters identified in the patient group and the controls and their analytical performances were evaluated (Table 5).

The distribution frequency of the ten clusters, considered both individually and cumulatively, is significantly higher in the patient group than in the controls. On the other hand, as expected, the sensitivity calculated for each cluster in the identification of allergic subjects is poor, not exceeding 50% even when the cumulative effect of the 10 clusters is considered.

On the contrary the specificity and positive predicting values are upper than 95 and 90%, respectively, in all cases. Together these data indicate that the clusters of SNPs identified by the a priori algorithm potentially have a high probability of predicting the genetic predisposition to airway allergic diseases.

## 4. Discussion

The genetic predisposition to allergic diseases, in particular those affecting both high and low airways, characterizes atopic condition and generally is defined as the familial genetic background favoring allergic disease development [34]. The genetic traits of allergy, however, are still not well delineated and studies, currently developed with different approaches such as genome-wide or candidate gene, linkage analysis and association, as well as functional studies of gene–gene or signaling pathways interaction, did not produce univocal results [35]. Generally, polymorphisms of Th2 cytokine genes have been found associated with atopy [13]. In particular, IL-4 rs2243250, IL-4R rs1801275, and IL-13 rs1800925 variants, affecting expression and/or functionality of the gene products [17,18,22] were considered markers of predisposition to allergy [22,36,37,38,39,40,41,42]. We were unable to demonstrate an association of IL-4 rs2243250 and IL-4R rs1801275 with airway allergic diseases in our population. On the other hand, the IL-13 rs1800925CC genotype was found to be associated with moderate risk for allergy (OR: 1.29, 95%CI: 1.11–1.51) whereas the presence of IL-13 rs1800925T positive genotypes seems to be protective (heterozygous genotype OR: 0.58 95%CI: 0.40–0.84; homozygous genotype OR: 0.42 95%CI: 0.19–0.89). The apparent discrepancies between our data and that reported in the literature were overcome when the a priori algorithm methodology was applied (see above). However, we do have not a simple explanation for our results. Considering the relatively reduced odd ratio value obtained by analyzing IL-13 rs1800925 genotype distribution we can hypothesize that the contribution of a single risk genotype in multifactorial diseases such as airway allergic diseases might be not detected using a classical statistic approach.

A crucial molecule involved in asthma and other airway allergies is the Type II receptor for IgE (FcεRII) known as CD23. CD23 is expressed in different cell types. All cells involved in immune response (T cells, B cells, and effector cells) and epithelial lining of airways express CD23 and the receptor participates in transcytosis of IgE and/or IgE-antigen immune complexes (ICs) across human airway epithelial cells [43]. Our data indicated that the frequency of CD23 rs2228137T/T genotype, characterized by an enhanced functional activity of the receptor [25], is increased in allergic patients. The SNP is located in the extracellular domain of the molecule and modulates the normal control of cell function mediated by CD23 [25].

The role of IL-6 gene polymorphisms in allergic diseases appears to be relevant [44,45]. Accordingly, the increase in IL-6 rs1800795 GG genotype frequency, associated with increased production of IL-6 [19], in our allergic patient group seems to confirm data from previous studies of other groups [44,45] in populations of different ethnic backgrounds. In addition, functional studies have demonstrated that IL6 serum levels and their epigenetic markers might predict allergic disease severity [46,47].

We found that IL-10 rs1800896AA genotype frequency associated with reduced production of IL-10 [20] was increased in the allergic group. These findings seem to confirm reports by other groups [48]. Moreover, our data indicate that IL-10RB rs2834167G/* positive genotypes, associated with a higher mRNA and cell surface expression of IL-10Rβ receptor [21], are more frequent among allergic patients than controls. At this moment, to our knowledge, this is the first observation of the relationship among variants of IL-10RB and allergic diseases. On the other hand, IL-10RB is expressed on the airway epithelia and mediates the effects of the cytokines that regulate mucociliary barrier functions, such as interleukin-22 (IL-22) and interferon-λ (IFN-λ), both members of the IL-10 family [49]. IL-22 induces the production of antimicrobial peptides and upregulates junctional protein expression and IFN-λ, decreasing junctional integrity and facilitating immune cell migration [49]. In this complex and pleiotropic scenario, the genetically determined increased expression of IL-10Rβ might dysregulate cilia epithelium functionality during allergen-induced inflammation response.

Finally, the Klotho rs564481 TT genotype appears significantly increased in the allergic patient group. Although rs564481, located in exon 4, is a synonymous variant where a nucleotide transition from C to T resulted in no amino acid change, previous studies have demonstrated the possibility that this type of variation affects protein function, potentially through effects on mRNA stability and processing, translation kinetics, and protein folding [50].

The computational approach to biological systems has become increasingly frequent and the continued improvement of the methodologies that are rapidly leading to the introduction of artificial intelligence systems to obtain study models of complex biological systems and identify new diagnostic markers and molecular pathways that can also be targeted by innovative therapies. However, the number, size, and variability of computational models have become so complex that it is often difficult to reproduce simulation results and reuse [51]. In this view, we applied the so-called market basket analysis. This a priori algorithm was originally applied to the mining of a dataset containing purchases from a supermarket, highlighting general hidden trends that can arise from data constituted by a set of items bought by customers. More generally, the a priori algorithm is a well-established data mining technique that tries to find hidden structures in categorical data [16,33]. Our approach identified a set of rules that applied to an allergic or not allergic population that might allow for identifying allergy-predisposed people using, in the end, limited sets of cytokine gene polymorphisms. In particular, we obtained ten rules by which if the genotype background of an individual contains a gene cluster defined by one of the rules the probability that he might be classified as a patient affected by airway allergy is >90%.

To evaluate the potential usefulness of the methodology adopted, we measured the analytical performances (sensitivity, specificity, predictive positive and negative values) of the single rules and those of all 10 considered cumulatively. As expected, we found a good specificity and positive predictive value for each rule, but sensitivity and negative predictive values are not so satisfactory. In other terms, we should conclude that not all allergic subjects might be tagged by these rules however when the genetic background of the subject is characterized by the presence of one or more gene clusters identified by these rules the probability that he is non-allergic might be extremely low.

On the other hand, a limit of this paper is that the population sample size might be insufficient for a comprehensive analysis of multifactorial diseases like allergy, so in the future, an increase in the number of subjects studied, maybe through a multicenter collaboration, including studies of diverse ethnic origin population, would be necessary to validate the current data.

One of the advantages of the approach we used was to highlight the SNP clusters identified by the market basket analysis also the presence of genotypes of the SNPs IL-4 rs2243250 and IL-4R rs1801275 associated with allergy, respectively, in five and two of the clusters identified. Both SNPs have been studied by different groups and the association with allergic pathologies, in particular with asthma or rhinitis, seems to be fairly consolidated [18,38,39,52,53].

One other of the identified clusters consists of the simultaneous presence of the IL-10RB rs2834167G/G and IFNγR2 rs2834213G/G genotypes. The latter polymorphism was found associated with asthma in Caucasian populations [54]. The SNP is located in an intronic region and is in linkage disequilibrium with 3 other associated SNPs to high transcriptional activity in an “in vitro” model [55]. The role of IFNγR2 and more in general of Th1 cytokines in asthma appears complex. As reported by Raundhal et al., patients affected by severe asthma display a dominant TH1 signature and cytokine profiles almost different from the classical Th2 and administration of TH1 cytokines [4]. Moreover, there is mounting evidence that TH1 pathway gene products impinge on airway tissue pathological remodeling rather than on airway allergic disease susceptibility [8]. As discussed above, the genetic variant of IL-10RB might affect the mucociliar barrier functionality after the interaction with IL-22 or IFN-λ ligand [49]. Considering that IL-10RB rs2834167G/G genotype is a component of 9 out of 10 gene clusters evidenced by the a priori algorithm, we can hypothesize that our approach allowed for identifying in the same cluster both gene variants involved in susceptibility to allergy and gene variants affecting severity and duration of the disease.

In this view, we can also hypothesize that the presence of Klotho rs564481 and Klotho rs577912 SNPs in four of the ten gene clusters (two for each of the SNPs) might be associated with the role that αKlotho protein might play in attenuation of airway mucosa inflammation [56,57]. The Klotho rs577912CC genotype is associated with reduced production of αKlotho compared to genotypes positive for the A allele while as reported more on Klotho rs564481 TT genotype might influence at pretranscriptional level mRNA stability and processing.

## 5. Conclusions

In this paper, we have obtained data suggesting that the computational approach to the evaluation of gene variants potentially involved in airway allergy, allows us to evidence gene clusterizations that involve, not only classical genes associated with disease susceptibility but also variants of genes involved in airway structure/remodeling and regulation of lung function. The approach that we have used, even if limited by the number of SNP typed and by the relatively reduced number of subjects recruited, appears a potentially powerful tool to better delineate the complex genetic background that underlies airway allergic diseases, early identification of patients susceptible to airway allergic diseases, and discover novel potential therapeutic targets.

## Figures and Tables

**Figure 1 genes-15-01105-f001:**
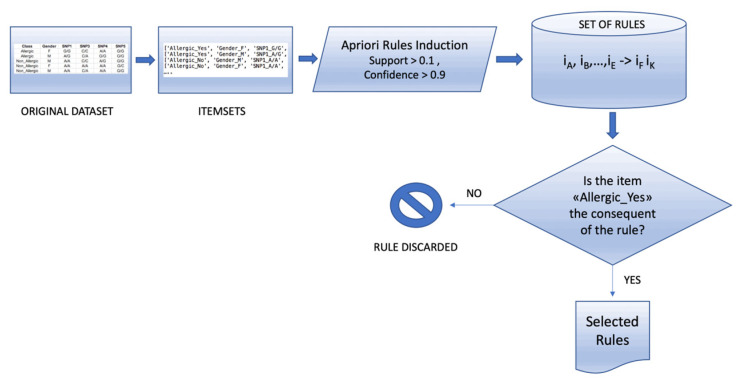
Flowchart of the market-basket analysis algorithm.

**Table 1 genes-15-01105-t001:** SNPs typed.

Gene	SNP	Chr Location	Major Allele	Minor Allele	Minor Allele Functional Association	Ref.
*IL-4*	rs2243250	5:132673462	C	T	Upregulation of gene expression	[17]
*IL-4R*	rs1801275	16:27363079	A	G	Enhanced IL-4 receptor function	[18]
*IL-6*	*rs1800795*	7:22727026	G	C	Reduced production of IL-6	[19]
*IL-10*	rs1800896	1:206773552	A	G	Increased production of IL-10	[20]
	rs1800872	1:206773062	C	A	Reduced production of IL-10	
*IL-10RB*	rs2834167	21:33268483	A	G	Higher expression of IL-10Rβ	[21]
*IL-13*	rs1800925	5:132657117	C	T	Increased production of IL-13	[22]
*IL-18*	rs187238	11:112164265	G	C	Increased IL-18 gene transcription	[23]
*IFNγ*	rs2430561	12:68158742	T	A	Lowered IFN-γ production	[24]
*IFNγR2*	rs2834213	21:3342060	A	G	Hyper responsiveness to Type II IFN	[21]
*CD23*	rs2228137	19:7698362	C	T	Increased activity of the receptor	[25]
*Klotho*	*rs577912*	13:33036014	C	A	Increased Klotho expression	[26,27]
	*rs564481*	13:33060846	C	T	mRNA stability modification	

**Table 2 genes-15-01105-t002:** Demographic and clinical characteristics of 308 patients with respiratory allergy and 205 control subjects.

Demographic and Clinical Characteristics	Allergy		Controls		*p*
	N.	%	N.	%	
Age, mean ± SD	51.03 ± 10.01		49.78 ± 12.32		0.227 ^a^
Women	167	54.22	117	57.07	0.527 ^b^
Clinical features of allergic patients					
Rhinitis	109	35.39	0	0	--
Rhinitis and asthma	93	30.19	0	0	--
Asthma	106	34.41	0	0	--
Positive familiarity	91	29.54	0	0	--
Positive prick test for ≥3 allergens	122	39.62	--	--	--
Atopy	172	55.84	0	0	--

^a^ Significant differences in age between the two groups was evaluated by Student *T* test. ^b^ Chi-square test was used to evaluate significant differences in gender distribution.

**Table 3 genes-15-01105-t003:** Allelic and genotypic frequencies of single nucleotide polymorphisms (SNPs) typed in a group of 308 patients affected by airway allergy and 205 matched controls.

GENE	SNP	Alleles/ Genotypes	Controls		Allergy		OR (95% CI)	^1^ *p*-Value
			Nr	Freq.	Nr	Freq.		
*IL-4*	*rs2243250*	C	363	0.89	560	0.91	1.29(0.86–1.95)	0.243
		T	47	0.11	56	0.09		
		C/C	162	0.79	259	0.84	1.40 (0.89–2.21)	0.159
		C/T	39	0.19	42	0.14	0.67 (0.42–1.08)	0.264
		T/T	4	0.02	7	0.02	1.17 (0.34–4.05)	1
*IL-4R*	*rs1801275*	A	330	0.8	507	0.82	1.13 (0.82–1.55)	0.461
		G	80	0.2	109	0.18		
		A/A	135	0.66	210	0.68	1.11 (0.76–1.62)	0.631
		A/G	60	0.29	87	0.28	0.93 (0.63–1.38)	0.733
		G/G	10	0.05	11	0.04	0.72 (0.30–1.73)	0.499
*IL-6*	*rs1800795*	G	278	0.68	484	0.79	1.74 (1.31–2.31)	0.0001
		C	132	0.32	132	0.21		
		G/G	98	0.48	201	0.65	2.05 (1.43–2.94)	0.0001
		G/C	82	0.4	82	0.27	0.49 (0.33–0.72)	0.0019
		C/C	25	0.12	25	0.08	0.64 (0.35–1.14)	0.132
*IL-10*	*rs1800896*	A	259	0.63	426	0.69	1.31 (1.00–1.70)	0.049
		G	151	0.37	190	0.31		
		A/A	79	0.38	147	0.48	1.46 (1.02–2.09)	0.046
		A/G	101	0.49	132	0.43	0.70 (0.48–1.02)	0.114
		G/G	25	0.12	29	0.09	0.63 (0.34–1.14)	0.378
	*rs1800872*	C	292	0.71	424	0.69	0.89 (0.68–1.17)	0.445
		A	118	0.29	192	0.31		
		C/C	101	0.49	144	0.47	0.90 (0.64.1.29)	0.589
		C/A	90	0.44	136	0.44	1.06 (0.73–1.53)	0.623
		A/A	14	0.07	28	0.09	1.40 (0.70–2.80)	0.413
*IL-10RB*	*rs2834167*	A	301	0.73	285	0.46	0.31 (0.24–0.41)	<0.0001
		G	109	0.27	331	0.54		
		A/A	108	0.53	79	0.26	0.49 (0.39–0.61)	<0.0001
		A/G	85	0.41	127	0.41	2.06 (1.38–3.07)	<0.0001
		G/G	12	0.06	102	0.33	5.66 (3.19–10.02)	<0.0001
*IL-13*	*rs1800925*	C	294	0.72	502	0.81	1.74 (1.29–2.34)	0.0003
		T	116	0.28	114	0.19		
		C/C	105	0.51	204	0.66	1.29 (1.11–1.51)	0.0009
		C/T	84	0.41	94	0.3	0.58 (0.40–0.84)	0.0013
		T/T	16	0.08	10	0.03	0.42 (0.19–0.89)	0.024
*IL-18*	*rs187238*	G	284	0.69	457	0.74	1.27 (0.97–1.68)	0.088
		C	126	0.31	159	0.26		
		G/G	100	0.49	170	0.55	1.13 (0.95–1.35)	0.176
		G/C	84	0.41	117	0.38	0.82 (0.56–1.19)	0.231
		C/C	21	0.1	21	0.07	0.59 (0.31–1.14)	0.183
*IFNγ*	*rs24030561*	T	210	0.51	329	0.53	1.09 (0.85–1.40)	0.523
		A	200	0.49	287	0.47		
		T/T	52	0.25	86	0.28	1.10 (0.82–1.48)	0.543
		T/A	106	0.52	157	0.51	0.90 (0.59–1.37)	0.785
		A/A	47	0.23	65	0.21	0.84 (0.50–1.39)	0.663
*IFNγR*	*rs2834213*	A	249	0.61	403	0.65	1.22 (0.94–1.58)	0.128
		G	161	0.39	213	0.35		
		A/A	71	0.35	119	0.39	1.12 (0.88–1.41)	0.401
		A/G	107	0.52	165	0.54	0.92 (0.63–1.34)	0.135
		G/G	27	0.13	24	0.08	0.53 (0.28–0.99)	0.051
*CD23*	*rs2228137*	C	324	0.79	436	0.71	0.64 (0.48–0.86)	0.0036
		T	86	0.21	180	0.29		
		C/C	125	0.61	181	0.59	0.96 (0.83–1.11)	0.647
		C/T	74	0.36	74	0.24	0.67 (0.51–0.87)	0.0039
		T/T	6	0.03	53	0.17	6.15 (2.57–14.76)	<0.0001
*Klotho*	*rs577912*	C	356	0.87	525	0.85	0.87 (0.61–1.26)	0.522
		A	54	0.13	91	0.15		
		C/C	154	0.75	221	0.72	0.95 (0.86–1.06)	0.418
		C/A	48	0.23	83	0.27	1.21 (0.80–1.82)	0.654
		A/A	3	0.02	4	0.01	0.90 (0.20–4.09)	1
	*rs564481*	T	145	0.35	376	0.61	2.86 (2.21–3.71)	<0.0001
		C	265	0.65	240	0.39		
		T/T	25	0.12	122	0.4	3.25 (2.19–4.81)	<0.0001
		C/T	95	0.46	132	0.43	0.28 (0.17–0.47)	<0.0001
		C/C	85	0.42	54	0.18	0.42 (0.32–0.57)	<0.0001

Table reports number of subjects (Nr) positive for each allele and genotype typed and the relative allelic or genotypic frequency (Freq.). OR = odds ratio; 95% CI = 95% of confidence interval. ^1^: Fisher exact test.

**Table 4 genes-15-01105-t004:** Rules identifying allergic people (support >0.1; confidence ≥0.9).

Rule N.	Rules	Conf	Supp	Li	Conv
Allergy 01	IFNγR2 rs2834213A/G; IL-10RB rs2834167G/G	0.934	0.111	1.556	6.094
Allergy 02	Klotho rs577912C/C; IL-10RB rs2834167G/G	0.900	0.140	1.499	3.996
Allergy 03	IL-10 rs1800896A/A; IL-10RB rs2834167G/G	0.917	0.107	1.527	4.795
Allergy 04	IL-10RB rs2834167G/G; IL-4 rs2243250C/C	0.901	0.160	1.501	4.041
Allergy 05	IL-10RB rs2834167G/G; IL-4R rs1801275A/A	0.903	0.127	1.504	4.110
Allergy 06	Klotho rs577912C/C; IL-10RB rs2834167G/G; IL-4 rs2243250C/C	0.906	0.113	1.509	4.263
Allergy 07	Klotho rs564481T/T; IL-6 rs1800795G/G; IL-13 rs1800925C/C	0.941	0.125	1.568	6.793
Allergy 08	IL-10RB rs2834167G/G; IL-6 rs1800795G/G; IL-4 rs2243250C/C	0.906	0.113	1.509	4.263
Allergy 09	IL-10RB rs2834167G/G; IL-4 rs2243250C/C; IL-4R rs1801275A/A	0.912	0.101	1.519	4.556
Allergy 10	Klotho rs564481T/T; IL-6 rs1800795G/G; IL-4 rs2243250C/C; IL-13 rs1800925C/C	0.930	0.103	1.549	5.694

Conf: confidence; Supp: support; Li: lift; Conv: conviction.

**Table 5 genes-15-01105-t005:** Performances of the rules identifying allergic people.

Rule		Allergic Subjects		Control Subjects		OR (95% C.I)	*p*	Sens.	Spec.	PPV	NPV
		N.	%	N.	%						
Allergy 01	Pos	57	18.51	4	1.95	11.41 (4.07–31.99)	0.0001	0.185	0.985	0.934	0.445
	Neg	251	81.49	201	98.05						
Allergy 02	Pos	72	23.37	8	3.90	12.10 (4.97–30.76)	<0.0001	0.234	0.961	0.900	0.455
	Neg	236	76.63	197	96.10						
Allergy 03	Pos	55	17.86	5	2.44	8.70 (3.42–22.1)	<0.0001	0.179	0.976	0.917	0.445
	Neg	253	82.14	200	97.56						
Allergy 04	Pos	82	26.62	9	4.39	7.90 (3.87–16.14)	<0.0001	0.266	0.956	0.901	0.464
	Neg	226	73.38	196	95.61						
Allergy 05	Pos	65	21.10	7	3.41	7.57 (3.39–16.87)	<0.0001	0.211	0.966	0.903	0.449
	Neg	243	79.90	198	96.59						
Allergy 06	Pos	58	18.83	6	2.91	7.69 (3.25–18.20)	<0.0001	0.188	0.971	0.906	0.453
	Neg	250	81.17	199	97.09						
Allergy 07	Pos	64	20.78	4	1.95	13.18 (4.72–36.83)	<0.0001	0.208	0.981	0.941	0.452
	Neg	244	79.22	201	98.05						
Allergy 08	Pos	58	18.83	6	2.91	7.69 (3.25–18.20)	<0.0001	0.188	0.971	0.906	0.443
	Neg	250	81.17	199	97.09						
Allergy 09	Pos	52	16.88	5	2.44	8.12 (3.18–20.72)	<0.0001	0.169	0.976	0.912	0.439
	Neg	256	83.12	200	97.56						
Allergy 10	Pos	53	17.21	4	1.95	10.44 3.72–29.35)	<0.0001	0.172	0.981	0.929	0.441
	Neg	255	82.79	201	98.05						
All. RULES	Pos	144	46.75	14	6.83	11.98 (6.66–21.55)	<0.0001	0.468	0.932	0.911	0.538
	Neg	164	53.25	191	93.17						

Pos: Subjects bearing the gene cluster identified by the rule; Neg: Subjects not bearing the gene cluster identified by the rule; OR: odds ratio; 95% CI: 95% confidence interval; Sens.: sensitivity; Spec.: specificity; PPV: positive predicting value; NPV: negative predicting value.

## Data Availability

All data generated or analyzed during this study are stored in electronic archives that can be supplied on request.

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
