# Peer review of "A Heuristic Approach to Analysis of the Genetic Susceptibility Profile in Patients Affected by Airway Allergies"

_genes, 2024, doi:10.3390/genes15081105_

Round 1
Reviewer 1 Report
Comments and Suggestions for Authors
This manuscript describes a study using a heuristic approach to identify genetic susceptibility profiles for allergic respiratory diseases like asthma, where genetic factors are likely major contributors. The researchers analyzed 13 SNPs related to cytokine and receptor genes (e.g., IL-4, IL-6, IL-13, IFNγ) in 308 allergic patients and 205 controls. They published findings on 10 genotype patterns linked to allergy susceptibility. These results suggest that data-mining approaches can identify predictive and diagnostic molecular markers, offering potential new therapeutic targets. This study is innovative in its simultaneous analysis of multiple genes and their interactions. It has significant public health implications, as many prevalent diseases are also influenced by multiple genes. The study's methods can be applied to investigate these diseases, providing critical information for disease management and prevention. While the manuscript is generally well-written, some errors need correction before publication. The detailed suggestions are as follows:
Some of your sentences closely resemble those from other publications. Please revise them to ensure originality, as they may otherwise be considered plagiarism. Please refer to the attached Similarity Report, where the highlighted sentences indicate the areas of concern.
From lines 57 to 61, the expression is awkward. Please have an expert in English scientific writing edit it to enhance the logical flow and clarity.
From lines 63 to 72, the detailed explanation is unnecessary, as it does not directly relate to your research topic on genetic susceptibility in airway allergy patients.
The Introduction section is lengthy; please shorten it to half its current length by removing content not directly related to your research topic.
From lines 90 to 93, and lines 93 to 95, you need citations for these statements.
The sentence from lines 104 to 107 is lengthy and unclear. Scientific writing prioritizes clarity over sentence length. This sentence is also redundant, as the same meaning is conveyed in the previous paragraph. Please state your research objective clearly at the end of the introduction section.
From lines 146 to 147, please report the accuracy and reliability of the SDS software used for revealing genotypes. Please do the same to the method of allele-specific PCR assays.
From lines 151 to 153, please provide the accuracy and reliability of the online statistical analysis tool used for evaluating SNP allele and genotype frequencies. Also, please explain the power of calculation in the same manner.
In line 153, please explain what Hardy-Weinberg equilibrium is and why it needs to be tested, as well as the power of calculation.
From lines 157 to 160, please explain how you built the multiple logistic regression models, including how covariates were selected. Please also clarify what the dominant and recessive models are, how they were defined, and why.
From lines 162 to 165, there is a grammar error; it appears a period is missing. Please also clarify whose performances (sensitivity, specificity, positive predictive value, negative predictive value) the software is evaluating. If related to the Heuristic analysis, please discuss it in the Heuristic Analysis section.
The Heuristic Analysis section lacks an explanation of the purpose of each step and how these relate to your research objectives. For example, please clarify the purpose of encoding items as “Feature Name Feature Value,” and please define the items I and database D, as well as X and Y in your study population. Please also explain the meaning of the rules and the four measures of the rules in the context of your study.
Table 3 contains abbreviations, so please include footnotes explaining them. The title of Table 3 is a conclusion and should be revised to reflect the table's descriptive content, not its implications.
From lines 215 to 222, these statements are not appropriate for the Results section and should be moved to the Methods section. However, avoid discussing results in the Methods section, so please edit accordingly to remove any discussion of results.
Lines 230 to 235 contain grammar errors, reducing clarity and coherence. Please have an expert in English scientific writing edit this paragraph.
A citation is missing in line 261.
The format of the Discussion section lacks consistency. Avoid using definitive terms like "demonstrating"; instead, use "showing," as scientific studies indicate probabilities rather than a 100% certainty.

The submitted manuscript is generally well-written and professional, but it contains an issue with unnecessarily lengthy sentences. The authors often combine several ideas into one long sentence, resulting in a loss of clarity and awkward phrasing. This issue is present throughout the manuscript and impacts its scientific clarity. Therefore, I recommend having an expert in English scientific writing edit the manuscript to improve clarity and logical flow.
Author Response
Referee 1
This manuscript describes a study using a heuristic approach to identify genetic susceptibility profiles for allergic respiratory diseases like asthma, where genetic factors are likely major contributors. The researchers analyzed 13 SNPs related to cytokine and receptor genes (e.g., IL-4, IL-6, IL-13, IFNγ) in 308 allergic patients and 205 controls. They published findings on 10 genotype patterns linked to allergy susceptibility. These results suggest that data-mining approaches can identify predictive and diagnostic molecular markers, offering potential new therapeutic targets. This study is innovative in its simultaneous analysis of multiple genes and their interactions. It has significant public health implications, as many prevalent diseases are also influenced by multiple genes. The study's methods can be applied to investigate these diseases, providing critical information for disease management and prevention. While the manuscript is generally well-written, some errors need correction before publication.
We thank referee for the general positive comments on the paper. His detailed suggestions were really helpful in improving manuscript quality. As suggested manuscript was revised by an expert in English scientific writing and edited accordingly.
- Some of your sentences closely resemble those from other publications. Please revise them to ensure originality, as they may otherwise be considered plagiarism. Please refer to the attached Similarity Report, where the highlighted sentences indicate the areas of concern.
Thank you for the advice. Using the similarity report that the referee kindly provided us, we have modified sentences with similarity. Of course commonly used scientific definitions as e.g. “single nucleotide polymorphisms”, “interleukins”, “genetic predisposition” or mathematical formulas and similar have not considered for changes
- From lines 57 to 61, the expression is awkward. Please have an expert in English scientific writing edit it to enhance the logical flow and clarity.
Sentences were rewritten according to referee suggestion
- From lines 63 to 72, the detailed explanation is unnecessary, as it does not directly relate to your research topic on genetic susceptibility in airway allergy patients.
Accordingly to referee suggestion, sentence has been simplified
- The Introduction section is lengthy; please shorten it to half its current length by removing content not directly related to your research topic.
Introduction was rewritten following referee suggestion
- From lines 90 to 93, and lines 93 to 95, you need citations for these statements.
Following the opportune referee suggestion, the necessary citations were added
- The sentence from lines 104 to 107 is lengthy and unclear. Scientific writing prioritizes clarity over sentence length. This sentence is also redundant, as the same meaning is conveyed in the previous paragraph. Please state your research objective clearly at the end of the introduction section.
we agree with the referee's observation, so the sentences at the end of the introduction were modified accordingly
- From lines 146 to 147, please report the accuracy and reliability of the SDS software used for revealing genotypes. Please do the same to the method of allele-specific PCR assays.
Following referee suggestion analytic performances of the SDS software and of KASPar assay were referred to that declared by manufacturers
- From lines 151 to 153, please provide the accuracy and reliability of the online statistical analysis tool used for evaluating SNP allele and genotype frequencies. Also, please explain the power of calculation in the same manner.
According to referee suggestions a brief description of accuracy and reliability of the online statistical analysis tool and of the power of calculation have been included in the revised manuscript as well as the relative references
- In line 153, please explain what Hardy-Weinberg equilibrium is and why it needs to be tested, as well as the power of calculation.
According to referee suggestions meaning and importance of the Hardy-Weinberg equilibrium and of the power of calculation was included
- From lines 157 to 160, please explain how you built the multiple logistic regression models, including how covariates were selected. Please also clarify what the dominant and recessive models are, how they were defined, and why.
We are indebted with the referee for this observation. In the original manuscript we repeat, by error, the description of the dominant model for recessive model. Error was corrected and a clearer description of logistic regression criteria was provided
- From lines 162 to 165, there is a grammar error; it appears a period is missing. Please also clarify whose performances (sensitivity, specificity, positive predictive value, negative predictive value) the software is evaluating. If related to the Heuristic analysis, please discuss it in the Heuristic Analysis section.
According to referee suggestion sentences were almost completely rewritten and the method to evaluate the potential diagnostic and predictive performaces (sensitivity, specificity, positive predictive value, negative predictive value) of the rules identified by the Heuristic analysis inserted at the end of this section
- The Heuristic Analysis section lacks an explanation of the purpose of each step and how these relate to your research objectives. For example, please clarify the purpose of encoding items as “Feature Name Feature Value,” and please define the items I and database D, as well as X and Y in your study population. Please also explain the meaning of the rules and the four measures of the rules in the context of your study.
Thank you for your suggestions: we have rewritten the section and improved the description of the mapping of data from the dataset to the itemsets list. In addition, we have inserted a graphical representation of the procedure to facilitate the understanding of the process by all readers.
- Table 3 contains abbreviations, so please include footnotes explaining them. The title of Table 3 is a conclusion and should be revised to reflect the table's descriptive content, not its implications.
We thanks referee for the helpful comment. The original Table 3 contained only data on SNP with a significant associations with allergy. The complete data of all SNPs were included in supplementary table S1. To satisfy referee request we have abolished table S1 and inserted in table 3 all SNP typing results, modifying accordingly the title. All abbreviations were explained in the footnotes.
- From lines 215 to 222, these statements are not appropriate for the Results section and should be moved to the Methods section. However, avoid discussing results in the Methods section, so please edit accordingly to remove any discussion of results.
We thanks referee for this observation. We have modified the sentence of the results section and, as reported above, we have we rewritten the Heuristic Analysis section
- Lines 230 to 235 contain grammar errors, reducing clarity and coherence. Please have an expert in English scientific writing edit this paragraph.
We thanks the referee for the advice, sentence has been rewritten accordingly
- A citation is missing in line 261.
Thank you for the advice. The reference has been added in the revised manuscript
- The format of the Discussion section lacks consistency. Avoid using definitive terms like "demonstrating"; instead, use "showing," as scientific studies indicate probabilities rather than a 100% certainty.
Discussion have been reshaped following referee’ suggestions
- The submitted manuscript is generally well-written and professional, but it contains an issue with unnecessarily lengthy sentences. The authors often combine several ideas into one long sentence, resulting in a loss of clarity and awkward phrasing. This issue is present throughout the manuscript and impacts its scientific clarity. Therefore, I recommend having an expert in English scientific writing edit the manuscript to improve clarity and logical flow.
Accordingly to referee’s suggestion manuscript was revised by an expert in English scientific writing.
Additional changes of the paper have been made according to the suggestions of other referees.

Reviewer 2 Report
Comments and Suggestions for Authors
The Authors applied association rule mining for SNP for explaining complexity of allergic respiratory diseases.
The paper is interesting and ell written but I also has some problems:
1. How was the size of the research sample determined? This must be clearly stated.
2. Page 3, Table 1, lines 131-132. It is reported p=0.227 and p=0.527. What test was used?
3. Page 4, line 153, "Power of calculation have been evaluated for all comparisons..." It should be reported this power.
It is for one comparison of for few comparisions?
4. Page 5, lines 168-170. "The features considered for each file, i.e., for each classification task, are: gender, age and the thirteen
SNPs typed." Do you mean logistic regression or another classifier?
5. Page 5, In 2.4. Heuristic Analysis. line 173, "each item is encoded as "Feature Name Feature Value" " What it means?
Maybe that type of SNPs are items (transactions in market basked analysis?). Give more explanation about it.

Author Response
Referee 2
The Authors applied association rule mining for SNP for explaining complexity of allergic respiratory diseases.
The paper is interesting and ell written but I also has some problems:
We thank referee for the general positive comments on the paper.
- How was the size of the research sample determined? This must be clearly stated.
We tanks the referee for the comment. As reported in the material and methods section we recruited allergic subjects and controls from March 2015 to December 2018, then we have calculated the minimum power of calculation for both groups as follows, to establish the sample size necessary to identify a real difference among the groups from that obtained by chance. This study had 80.0% power to detect a P1 = 0.066 with a Risk ratio = 0.220 and an odds ratio = 0.165 with a minimum sample size of 113 allergic subject and 75 control subjects for allergy. For the less frequent SNP genotype the minimum sample size was of 320 allergic subject and 190 control subjects. These data were now included in the material and methods section,
- Page 3, Table 1, lines 131-132. It is reported p=0.227 and p=0.527. What test was used?
We tanks the referee for the comment. Student T test for evaluate significant differences in age between the two groups and Chi square test for gender distribution. Explicative footnotes were added to the table 1
- Page 4, line 153, "Power of calculation have been evaluated for all comparisons..." It should be reported this power. It is for one comparison of for few comparisions?
According to referee suggestions we reported in the revised manuscript the power of calculation for the lowest expected frequency (calculated by Pearson’ test) among SNP genotypes analyzed, considering that higher genotype frequencies necessitate of a lower power of calculation. Values of the power of calculation and sample size were inserted in the revised manuscript
- Page 5, lines 168-170. "The features considered for each file, i.e., for each classification task, are: gender, age and the thirteen SNPs typed." Do you mean logistic regression or another classifier?
- Page 5, In 2.4. Heuristic Analysis. line 173, "each item is encoded as "Feature Name Feature Value" " What it means? Maybe that type of SNPs are items (transactions in market basked analysis?). Give more explanation about it.
Thank you for both these two criticisms: we have rewritten the section “2.4. Heuristic Analysis” and improved the description of the mapping of data from the dataset to the itemsets list. In addition, we have inserted a graphical representation of the procedure to facilitate the understanding of the process by all readers.
Additional changes of the paper have been made according to the suggestions of other referees.

Reviewer 3 Report
Comments and Suggestions for Authors
The manuscript presents an intriguing study on the genetic predisposition to airway allergies. The authors utilized a heuristic approach, specifically market basket analysis, to identify patterns of genetic polymorphisms associated with allergic susceptibility. The study is well-structured and provides valuable insights into the genetic factors contributing to airway allergies. However, some areas require clarification, additional detail, and minor corrections to enhance the manuscript's overall readability and impact.
1. Line 66:"trough"should be"through".
2.It might be better to clarify the term "heuristic approach" briefly in the Introduction for readers who may not be familiar with it.
3.Please provide more detail on the inclusion and exclusion criteria for patient and control selection to ensure reproducibility. Discuss potential confounders and how they were controlled in the analysis.
4.The selection of SNPs is well-explained. However, a brief justification for choosing these specific SNPs over others would be helpful.
5.The"2.4 Heuristic Analysis"section might be more straightforward if it was presented graphically.
6.Table 3 should have a clear explaining all abbreviations used.
7.Table 4 Allergy 08"rs1800795_G/G"should be"rs1800795G/G".
8.Line 233-235" IL-13 rs1800925C/C in three, IL-6 rs1800795G/G in two Klotho rs577912C/C and in another two Klotho rs564481T/T. finally one of the clusters contains IFNγR2 rs2834213G/G genotype"need minor corrections.It should be" IL-13 rs1800925C/C in three, IL-6 rs1800795G/G in two,Klotho rs577912C/C and another Klotho rs564481T/T in two respectively,finally one of the clusters contains IFNγR2 rs2834213G/G genotype."
9. The controls group needs to perform a series of clinical examinations as the patients group, which requires a substantial energy and time as well as funding, but the manuscript declares that this research received no external funding, so how to ensure the quality of the completion and the compliance of the population.
10.The sample size, though moderate, is insufficient for a comprehensive analysis of multifactorial diseases like allergy,results in a low number of some clusters in Tables 3 and 5 as well as limits the generalizability of the findings.It is suggested to increase the sample size in the future, and even worldwide collaborations, including studies of diverse races to validate the current conclusions are more convincing.

No problem.
Author Response
Referee 3
The manuscript presents an intriguing study on the genetic predisposition to airway allergies. The authors utilized a heuristic approach, specifically market basket analysis, to identify patterns of genetic polymorphisms associated with allergic susceptibility. The study is well-structured and provides valuable insights into the genetic factors contributing to airway allergies. However, some areas require clarification, additional detail, and minor corrections to enhance the manuscript's overall readability and impact.
We thank referee for the general positive comments on the paper. His detailed suggestions were really helpful in improving manuscript quality.
- Line 66:"trough"should be"through".
Thank you for your advice. However following the suggestions of another referee the sentence was deleted
2.It might be better to clarify the term "heuristic approach" briefly in the Introduction for readers who may not be familiar with it.
Thank you for the comment. A brief explanation of “what heuristic analysis is” was now inserted in the introduction
3.Please provide more detail on the inclusion and exclusion criteria for patient and control selection to ensure reproducibility. Discuss potential confounders and how they were controlled in the analysis.
According to referee suggestions we have better detailed the recruitment criteria and subject evaluation procedures in material and methods section.
4.The selection of SNPs is well-explained. However, a brief justification for choosing these specific SNPs over others would be helpful.
We thanks the referee for the suggestion. Each SNP was selected on the basis of the possible impact of the functional modification of the gene products induced by the SNPS to facilitate an immediate reading of characteristics of each SNP table 2 (now table 1) was inserted at the end of the introducion
5.The"2.4 Heuristic Analysis"section might be more straightforward if it was presented graphically.
We are indebted with the referee for the suggestion: we have rewritten the section and improved the description of the method. Accordingly to the suggestion, we have inserted a graphical representation of the procedure to facilitate the understanding of the process by all readers.
6.Table 3 should have a clear explaining all abbreviations used.
We thanks referee for the helpful comment. In the revised manuscript, all abbreviations of table 3 were explained in the footnotes.
7.Table 4 Allergy 08"rs1800795_G/G"should be"rs1800795G/G".
Thank you, done
8.Line 233-235" IL-13 rs1800925C/C in three, IL-6 rs1800795G/G in two Klotho rs577912C/C and in another two Klotho rs564481T/T. finally one of the clusters contains IFNγR2 rs2834213G/G genotype"need minor corrections.It should be" IL-13 rs1800925C/C in three, IL-6 rs1800795G/G in two,Klotho rs577912C/C and another Klotho rs564481T/T in two respectively,finally one of the clusters contains IFNγR2 rs2834213G/G genotype."
We thanks the referee for the correction suggested, sentence has been rewritten accordingly
- The controls group needs to perform a series of clinical examinations as the patients group, which requires a substantial energy and time as well as funding, but the manuscript declares that this research received no external funding, so how to ensure the quality of the completion and the compliance of the population.
We thanks the referee for the suggestion. As above reported, we have better detailed the recruitment criteria and subject evaluation procedures in material and methods section. In particular we have based the control subjects recruitment on a general clinical examination and a thorough anamnesis and performed by the same allergology staff that evaluated the patients. We are aware that this method does not allow to exclude the presence of asymptomatic sensitizations towards some allergens. On the other hand, it is difficult, given the experience of the allergology staff involved, for an allergic subject to be completely asymptomatic.
10.The sample size, though moderate, is insufficient for a comprehensive analysis of multifactorial diseases like allergy,results in a low number of some clusters in Tables 3 and 5 as well as limits the generalizability of the findings.It is suggested to increase the sample size in the future, and even worldwide collaborations, including studies of diverse races to validate the current conclusions are more convincing.
We totally agree with this criticisms. We have inserted in discussion your observation as a limit of the paper and prospected further development of the research through collaboration with other research centers.
Additional changes of the paper have been made according to the suggestions of other referees.

Round 2
Reviewer 2 Report
Comments and Suggestions for Authors
I am satisfied, all ambiguities have been clarified.

Reviewer 3 Report
Comments and Suggestions for Authors
The manuscript has been modified and can be accepted.
Comments on the Quality of English LanguageNo problem.